# Selective signatures in composite MONTANA TROPICAL beef cattle reveal potential genomic regions for tropical adaptation

Camila Alves dos Santos[1]ᴼ*, Joanir Pereira Eler[2‡], Elisangela Chicaroni de Mattos Oliveira[2‡], Rafael Espigolan[2‡], Gabriela Giacomini[3‡], José Bento Sterman Ferraz[2‡], Tiago do Prado Paim[1]ᴼ

1 Programa de Pós-graduação em Zootecnia, Instituto Federal de Ciência, Educação e Tecnologia Goiano, Rio Verde, Goiás, Brazil, 2 Departamento de Zootecnia, Faculdade de Zootecnia e Engenharia de alimentos, Universidade de São Paulo, Pirassununga, São Paulo, Brazil, 3 Associação Internacional de criadores de Montana, Mogi Mirim, São Paulo, Brazil

ᴼ These authors contributed equally to this work.
‡ JPE, ECMO, RE, GG and JBSF also contributed equally to this work.
* camilaalvesdossantos240@gmail.com

**Data Availability Statement:** Data cannot be shared publicly because it belongs to the International Association of Montana Breeders

## Abstract

Genomic regions related to tropical adaptability are of paramount importance for animal breeding nowadays, especially in the context of global climate change. Moreover, understanding the genomic architecture of these regions may be very relevant for aiding breeding programs in choosing the best selection scheme for tropical adaptation and/or implementing a crossbreeding scheme. The composite MONTANA TROPICAL® population was developed by crossing cattle of four different biological types to improve production in harsh environments. Pedigree and genotype data (51962 SNPs) from 3215 MONTANA TROPICAL® cattle were used to i) characterize the population structure; ii) identify signatures of selection with complementary approaches, i.e. Integrated Haplotype Score (iHS) and Runs of Homozygosity (ROH); and iii) understand genes and traits related to each selected region. The population structure based on principal components had a weak relationship with the genetic contribution of the different biological types. Clustering analyses (ADMIXTURE) showed different clusters according to the number of generations within the composite population. Considering results of both selection signatures approaches, we identified only one consensus region on chromosome 20 (35399405–40329703 bp). Genes in this region are related to immune function, regulation of epithelial cell differentiation, and cell response to ionizing radiation. This region harbors the slick locus which is related to slick hair and epidermis anatomy, both of which are related to heat stress adaptation. Also, QTLs in this region were related to feed intake, milk yield, mastitis, reproduction, and slick hair coat. The signatures of selection detected here arose in a few generations after crossbreeding between contrasting breeds. Therefore, it shows how important this genomic region may be for these animals to thrive in tropical conditions. Further investigations on sequencing this region can identify candidate genes for animal breeding and/or gene editing to tackle the challenges of climate change.

(AIC-MTN). Data can be made available by the Montana Breeders Association (contato@montana.org.br) to researchers who meet their criteria for access to confidential data.

**Funding:** Instituto Federal de Educação, Ciência e Tecnologia Goiano (IF Goiano) and National Council for Scientific and Technological Development (CNPq). The funders had no role in study design, data collection and analysis, decision to publish, or preparation of the manuscript only in the financial support and the infrastructure.

**Competing interests:** The authors have declared that no competing interests exist.

# Background

Coping with environmental stress is a paramount physiological requirement for animals in subtropical and tropical regions [1]. The capacity to maintain a homeostatic body temperature in challenging environments allows animals to increase their production potential [2, 3]. Therefore, in a changing climate scenario, it is very important to understand the genetic background related to tropical adaptation.

Cattle are composed of two main subspecies: indicine or zebu (i.e., *Bos primigenius indicus*) and taurine (i.e., *Bos primigenius taurus*). Most of taurine animals were bred in temperate regions of the world [4]. As a result of the origins and breeding practices, both natural and artificial, cattle are broadly divided into temperate (taurine) and tropical (zebu) based on the common adaptation characteristics [5]. Notwithstanding, some taurine breeds adapted to tropical climate arose in some regions of Africa and America due to both natural and artificial breeding, e.g. Senepol [6] and Bonsmara [7].

These differences between biological groups can be used for crossbreeding to get animals better adapted to the environment and with higher production efficiency [8]. Kim et al. [9] showed that African cattle pastoralism succeeded due the arrival of indicine cattle and their crossbreeding with local taurine and that selection shaped the admixture proportion of taurine x indicine crossbreeds to increase diversity and facilitate evolutionary adaptation.

The composite MONTANA TROPICAL® cattle population (https://montana.org.br/) was developed in Brazil (starting in 1994) following the idea of using crossbreeding for formation of a tropical-efficient cattle population. This composite population (Fig 1) is based on crossbreeding of four different biological types or breed groups: 1) zebu breeds (N), 2) adapted taurine breeds (A), 3) British taurine breeds (B), and 4) Continental European taurine breeds (C) [10] (S1 Fig). The combination of these multiple breeds can change according to the farmer decisions based on

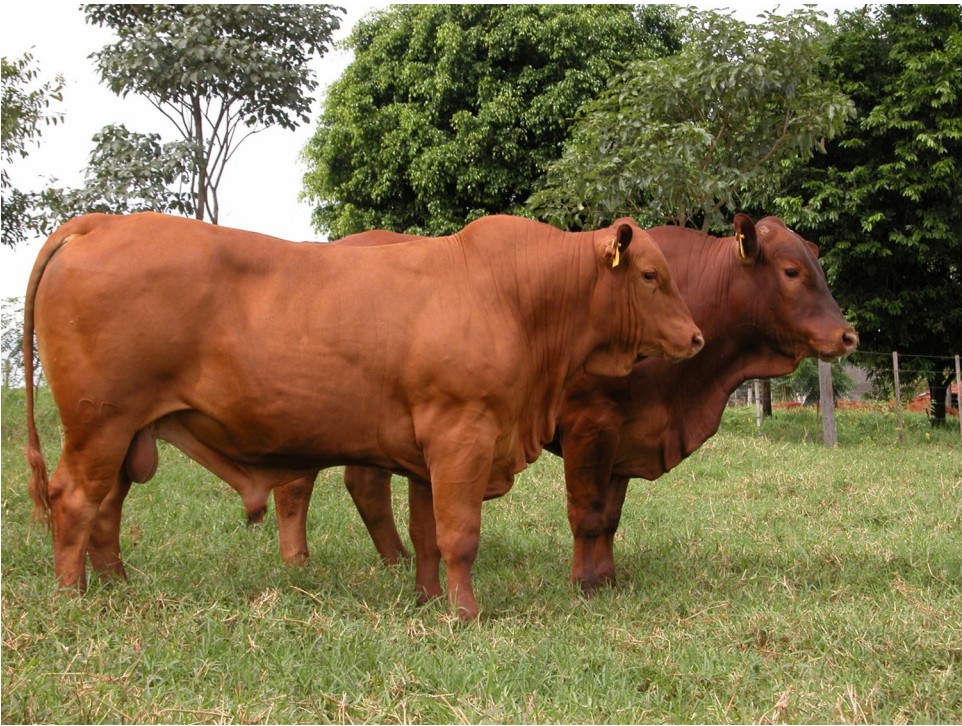

**Fig 1. Montana Tropical® animals.**

the farm environment, nevertheless animals must have at least three groups of NABC in their pedigree to be considered Montana Tropical® (https://montana.org.br/). The maximum proportion of each group allowed is 37.5% for group N; 87.5% for group A; and 75% for group B and C together [11, 12]. The breeding program of Montana Tropical® emphasize adaptation and robustness characteristics and growth traits account for 70% of their selection index [13].

Knowledge of population structure is important to animal breeding program. This knowledge allows breeders to make more assertive and accurate decisions in breeding, diversity preservation, and selection [14]. Genetic composition based on pedigree estimation methods do not account for Mendelian sampling. By using genomic data, estimation of genetic parameters is less subject to input pedigree errors and better accounts of Mendelian sampling [15].

Genomic data can be used to identify runs of homozygosity (ROH) which are continuous homozygous segments inherited from common ancestors [16,17]. The formation of ROH can be influenced by inbreeding, genetic drift, population bottlenecks, as well as natural and artificial selection. Moreover, the evaluation of ROH can characterize the history of the population structure [18].

By analyzing the size of the ROH, it is possible to calculate a genomic inbreeding coefficient and classify inbreeding, with long segments related to recent inbreeding and short ones to older inbreeding [19]. The population frequency of ROH can be used as a tool to identify genomic regions under selection (i.e. signature of selection) [20].

Another method to detect signals of selection is the Integrated haplotype score (iHS), which is a statistical approach based on extended haplotype homozygosity (EHH). This method detects positive selection by the increased mutation frequency and long-range linkage disequilibrium [21]. Ben-Jemaa et al. [22] used iHS and complementary methods to identify selection sweeps related to adaptation and immune response in Maremmana cattle.

To better identify signatures of selection it is a good strategy to combine complementary approaches. ROH and F$st$ methods, for example, were able to identify selected regions related to heat stress and immunity in tropically adapted breeds along with genes related to hypoxia factors in temperate breeds raised in low-oxygen environments [23]. Liu et al. [24], for example, used these tools to identify signatures of selection in Shanghai Holstein cattle.

Using the MONTANA TROPICAL® composite population, we sought in this work to (1) characterize the population structure of composite MONTANA TROPICAL® beef cattle population through generations inside the population; (2) identify signatures of selection with complementary approaches (ROH and iHS); and (3) characterize genes in significant genomic regions under selection.

## Results

### Population structure

The equivalent complete generation (ECG) is a value that indicates how many generations each animal has within the population pedigree. ECG in our data varies from 1 to 6, which agrees with the population history as it was formed 29 years ago. Principal component analysis (PCA) showed a weak relationship between ECG estimated by pedigree and the first two PCs (Fig 2). Animals with higher number of generation (>4 ECG) were located in positive side of PC1 and close to zero in PC2, while animals of the initial generations (<2 ECG) had three diversity spots, one in the negative side of PC1 and other two in opposite sides of PC2.

We conducted correlation and regression analyses to better comprehend the relationship between ECG and contribution ratio of the founder biological groups according to pedigree data (Zebu—N, Tropical Adapted Taurine–A, British Taurine–B and Continental Taurine—C). Tropical Adapted Taurine (A) and Continental Taurine (C) proportion had positive

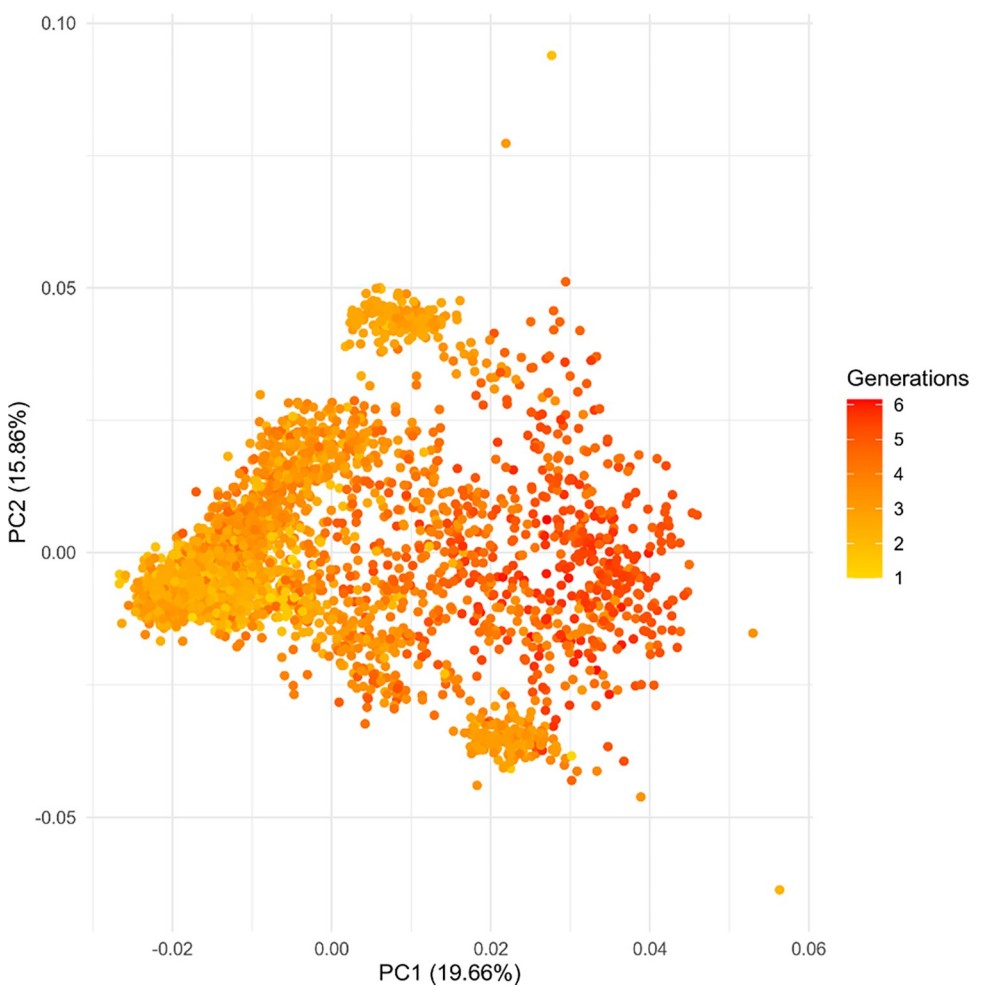

**Fig 2. First two components of principal component analysis using genomic data of 3215 MONTANA TROPICAL® animals.** Generations: corresponds to the number of ECG calculated based on pedigree data.

correlations (r = 0.33 and r = 0.16, respectively; p<0.001) with ECG. On the contrary, Zebu (N) and British Taurine (B) proportion had negative correlations (r = -0.19; and r = -0.29; p<0.001) with ECG. The regression analysis confirmed the trend of a 4% increase per generation in the proportion of Tropical Adapted Taurine (A) and a 3% decrease per generation in the proportion of British Taurine (B) based on pedigree within the composite population.

ADMIXTURE analyses demonstrated a uniform composition of recent MONTANA TROPICAL® animals (ECG≤2.5) at K = 2. Animals in intermediate generations (2.5<ECG≤4.5) had a more diverse composition. In advanced generations (ECG>4.5), animals with higher proportions of other clusters could be observed, differentiating this group of animals with the earlier generations of the population (Fig 3).

At K = 3, one new cluster (red in Fig 3) showed a higher frequency in animals from advanced generations. Therefore, the ADMIXTURE results clearly showed a different genomic composition of the animals through advancing ECGs within the MONTANA TROPICAL® population.

To explore the statistical relationship between ADMIXTURE clusters (K = 4) and ECG, we also performed correlation and regression analyses. ECG had positive correlations with Cluster

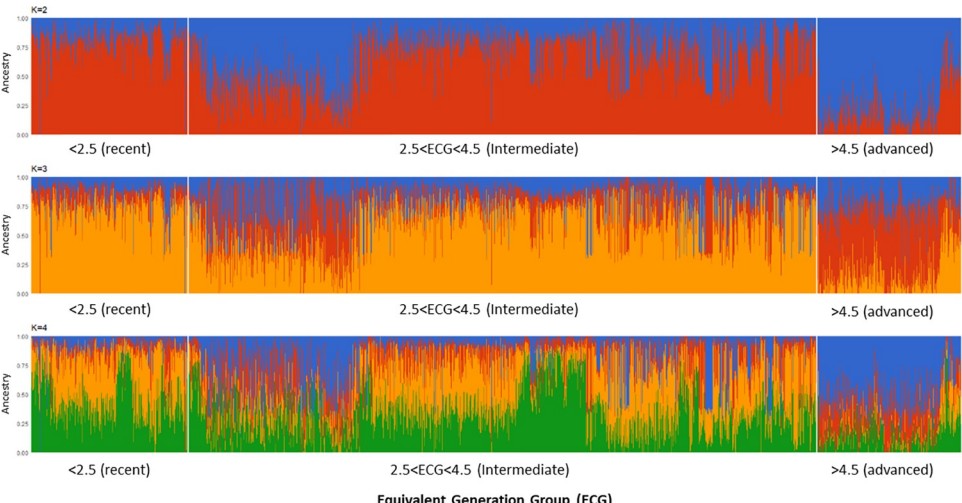

**Fig 3. Population structure of composite MONTANA TROPICAL® population inferred by ADMIXTURE software.** Each animal is represented by a single vertical bar divided into K colors, where K is the number of assumed ancestral groupings, which is plotted from K equal 2 to 4. Cluster definitions based on K = 4: Cluster 1 = Blue, Cluster 2 = Red, Cluster 3 = Orange and Cluster 4 = Green.

1 (r = 0.58, p<0.001) and Cluster 2 (r = 0.23, p<0.001). Clusters 3 and 4 had negative correlations with ECG (r = -0.45 and r = -0.42, respectively). The regression analyses showed trends of a: 12% increase of C1 per generation; 4% increase of C2 per generation; 9% decrease of C3 per generation; and 0.7% decrease of C4 per generation. Therefore, the statistical analyses confirmed the change in clustering of animals according to ECG, therefore it can be related to selection pressures working on shaping genomic structure of the composite population through generations.

We combined the data from the four first principal components (PC1 to PC4), the four clusters of ADMIXTURE (C1 to C4) and the NABC founder proportions from the pedigree to form a new dataset with 12 variables for each animal. This data was used to perform another PCA (Fig 4) to better understand the relationship between genomic structure of the population and founders' contribution ratios. We observed that PC1, ECG, C1, C2 and Tropical Adapted Taurine (A) proportion (based on pedigree) were all in a similar direction (Fig 4), suggesting that C1 and C2 represent the increase in genomic composition of Tropical Adapted Taurine (A) and the emergence of a genetic structure within MONTANA TROPICAL®. Cluster 3, PC3, and Zebu (N) proportion were also in the same direction (Fig 4). Cluster 4 and British Taurine (B) were also in the same position. Therefore, the ADMIXTURE clusters indicated the following associations: C1 with Tropical Adapted Taurine (A) and ECG; C2 with ECG; C3 with Zebu proportion (N); and C4 with British Taurine proportion (B).

### Signatures of selection

Of the 3215 evaluated animals, 3140 had at least one ROH segment in the genome, totaling 2349 ROH. The mean ROH length was 7.48 ± 0.08 Mb, varying until to 39 Mb (S1 Table). The average sum of all ROH segments per animal was 61.21 ± 1.10 Mb.

The genomic inbreeding coefficient ($F_{ROH}$) showed a significant positive regression with the inbreeding coefficient calculated using the pedigree (S2 Fig). The overall $F_{ROH}$ of the composite MONTANA TROPICAL® population was 2% while the pedigree-based inbreeding coefficient was 0.8%. There was an increase in $F_{ROH}$ as the number of ECGs increased (S3 Fig).

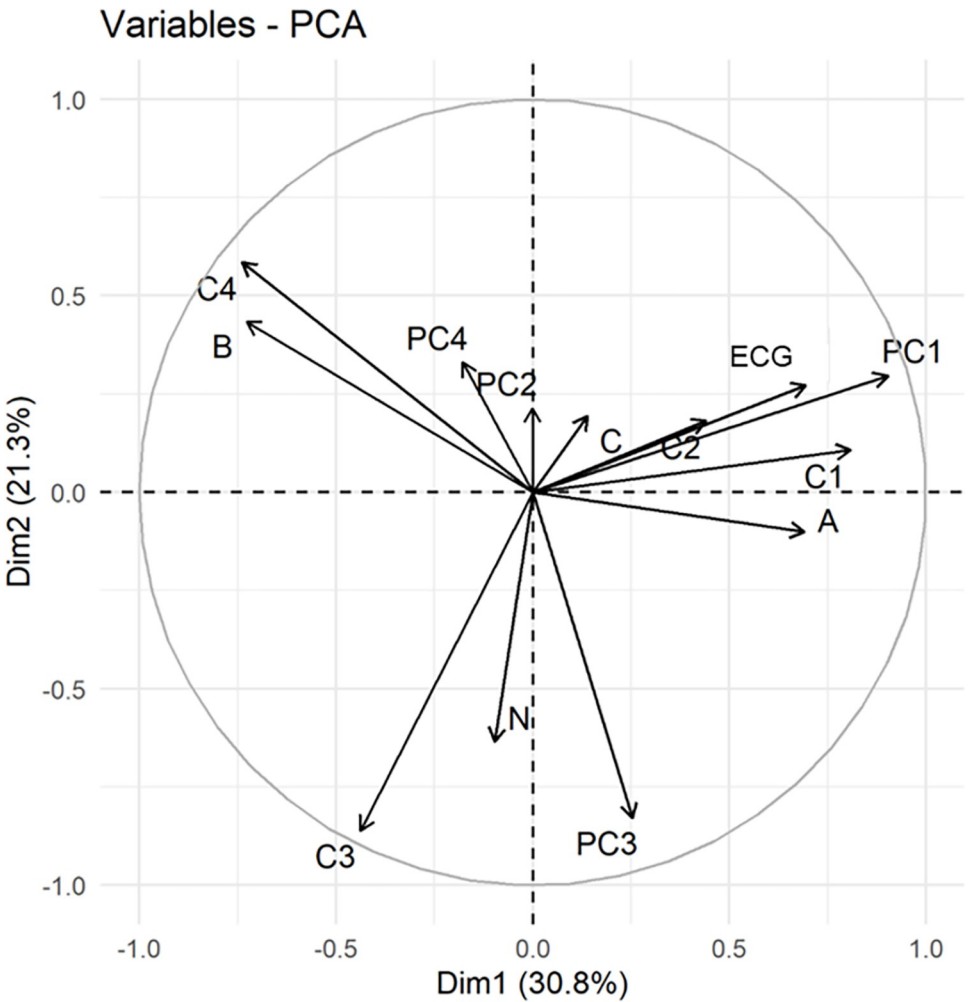

**Fig 4.** Results of the first two components of principal components analysis using the dataset formed by pedigree-based breed composition (N—Zebu, A–Adapted taurine, B–British taurine, C–European Continental taurine), four clusters of the ADMIXTURE analysis (C1, C2, C3, C4), equivalent complete generation (ECG), and the first four components of PCA analysis using genomic information (PC1, PC2, PC3, PC4).

There was no high-generation animal (ECG > 5) with $F_{ROH}$ equal to zero, demonstrating the accumulated inbreeding through generations inside the composite.

We classified ROH based on the length according to number of previous generations to the common ancestor (1Mb<ROH<6.5Mb = more than 6 generations ago, 6.5Mb<ROH<8.7 Mb = between 6 and 4.5, 8.7Mb<ROH<15.6Mb = between 4.5 and 2.5 generations ago, and ROH>15.6 Mb = less than 2.5 generations ago). In the population, 95.89% had some ROH of size 1Mb<ROH<6.5Mb. The ROH length representing a common ancestor within the previous 2.5 generations (> 15.6 Mb) were identified in 39.45% of the animals, indicating the emergence of very recent inbreeding. The advanced generations (ECG > 4.5) showed significantly higher values of ROH than all generations (p< 0.0001), especially for the largest ROH segments (S4 Fig).

Runs of homozygosity analyses identified six genomic regions as putative selected regions (BTAU 3, 5, 6, 14, 20, 21), using the 1% threshold. However, when using the top 0.1%, only one genomic region (BTAU 20) arises (Fig 5).

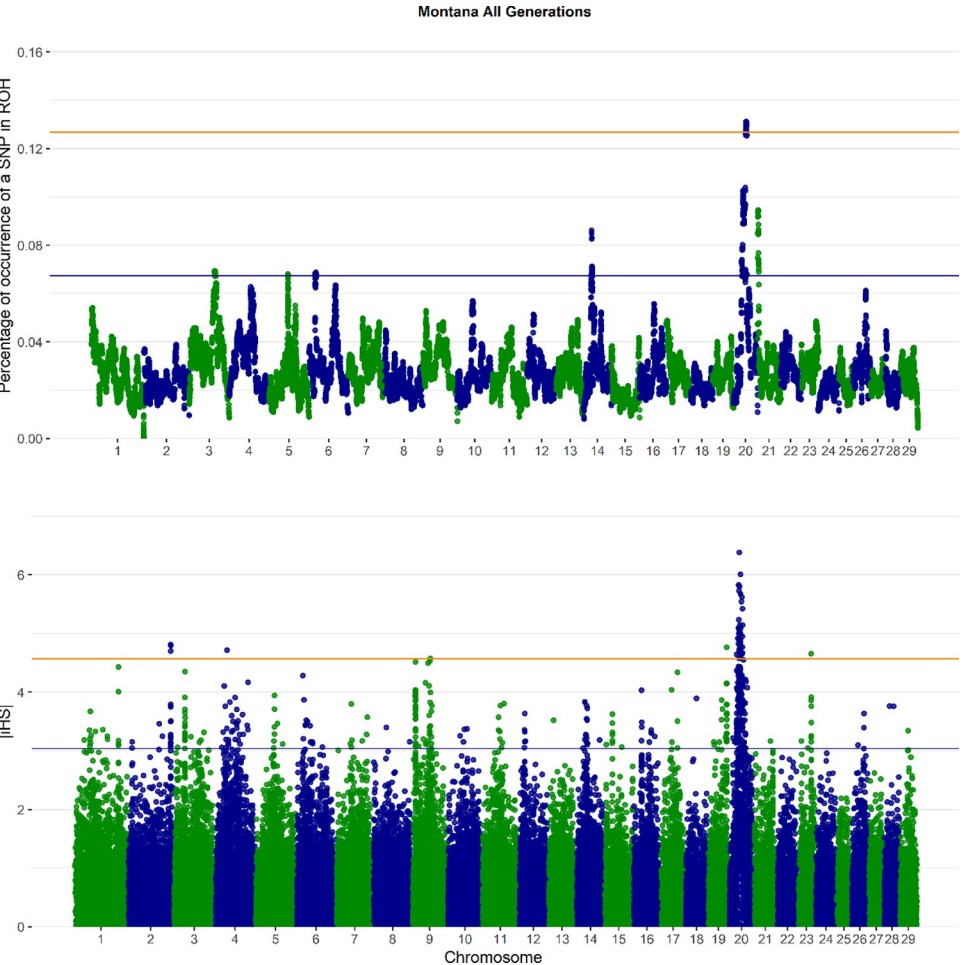

**Fig 5. Manhattan plot for iHS and ROH.** (A) Genome-wide distribution of signatures of selection detected by iHS. (B) Manhattan plot of incidence of each SNP in the ROH across individuals. The orange and blue line represents the threshold levels of top 1 % and 0.1%, respectively.

Identification of selection signatures was performed within three groups of animals based on ECG (<2.5 –recent; 2.5<ECG<4.5—intermediate; >4.5—advanced). The threshold of 1% (blue line) demonstrated different regions in each class of generation—recent (BTAU 3,5, 6, 9, 10, 13, 14, 21), intermediate (BTAU 3, 5, 14, 20, 21), advanced (BTAU 4, 6, 20, 26) (S5 Fig). The threshold of top 0.1% (orange line) showed ROH analysis found the same one significant region (BTAU 20) for the intermediate and advanced generation (S5 Fig), which was not seen on recent generation group. Moreover, the threshold of top 0.1% increased according to ECG, demonstrating the increasing frequency of homozygous segments according to selection through generations (S5 Fig).

The results of iHS analyses yielded estimated |iHS| scores for 46240 SNPs out of 51692 SNPs in total (S1 File). The 1% threshold (|iHS| > 2) showed several SNPs as possible signatures of selection. Most of chromosomes had some SNP above this threshold, of which only the BTAU 22, 24, 25 and 27 did not have a SNP with significant |iHS| score. When looking on the top 0.1% threshold, five regions were identified as putative selective signatures (BTAU 2, 4, 9,19, 20 and 21; Fig 4).

The selection signatures based on iHS (0.1% threshold) within ECG groups showed possible signatures of selection on: BTAU 1, 2, 9, 13, 15, 19, 20 e 23 for recent generations; BTAU 2, 9, 19, 20 and 23 for intermediate; and BTAU 1, 3 4, 6, 7, 9, 10, 12, 19, 20 and 21 for advanced generations. Therefore, the iHS results showed some variations between groups of animals.

## Consensus regions between ROH and iHS

Considering both analyses (ROH and iHS), there was a consensus region in chromosome 20 (BTAU 20). This region had a size equal to 4.9 Mb (20:35399405–40329703). In the ROH analysis, we found one signature of selection region within BTAU 20. Considering iHS, three regions of signatures of selection were observed within BTAU 20 (S2 Table).

When looking at the different ECGs some differences in the frequency of signatures of selection within BTAU20 were found. In the animals from ECG<2.5, there were no ROH within this region. In the intermediated generations (2.5<ECG<4.5), the frequency of SNPs within the ROH island on BTAU20 was around 10%. Meanwhile, in the advanced generations (ECG>4.5) the SNP frequency within the ROH island on BTAU20 was 30% (S5 Fig). The iHS analyses into the three ECG groups yield 3, 4, and 4 putative selection signatures within this region for recent (<2.5), intermediate (2.5<ECG<4.5) and advanced generations (>4.5), respectively (S4 Fig).

## Genes and QTLs in selected regions

The gene annotation under the consensus region of chromosome 20 found 32 genes. The QTLs were correlated with important features such as slick hair coat, feed intake, milk, reproduction, and immunity, which are summarized in Table 1.

To better understand the functions of the candidate genes, we uploaded the 32 genes to Cytoscape ClueGo and obtained the GO biological network (Fig 6A and 6B). More details of the network can be seen in S6 Fig. All these genes were found to be involved in several gene clusters (S2 File) and 12 GO terms were significantly enriched (Fig 6C).

## Discussion

### Population structure

The population structure of the composite MONTANA TROPICAL® population demonstrated through PCA showed a dispersion of the animals of more advanced generations according to PC1. ADMIXTURE results demonstrated that animals of ECG < 4.5 presented a different clustering when compared to earlier generations.

Principal components and ADMIXTURE clusters showed a weak relationship with the contribution ratio of the founder biological group. We expected a clearer relationship as some previous studies have used the results of PCA to calculate the proportion of each founder's breed

**Table 1. Genomic region identified by iHS and ROH analyses observed in MONTANA TROPICAL® composite animals, underlying genes and QTL identified in this region.**

| Chr | Start (Bp) | End (Bp) | Size | n Genes | Genes | QTLs |
|---|---|---|---|---|---|---|
| 20 | 35399405 | 40329703 | 4930298 | 32 | *RICTOR, OSMR, LIFR, EGFLAM, GDNF, WDR70, NIPBL, NUP155, MIR2360, CPLANE1, NIPBL, SLC1A3, RANBP3L, NADK2, SKP2, LMBRD2, UGT3A2, CAPSL, IL7R, SPEF2, PRLR, DNAJC21, BRIX1, RAD1, TTC23L, RAI14, C1QTNF3, AMACR, SLC45A2, RXFP3, ADAMTS12, TRNAT-UGU, TARS* | Slick hair coat, Milk, feed intake, immunology, mastitis, meat, reproduction, weight |

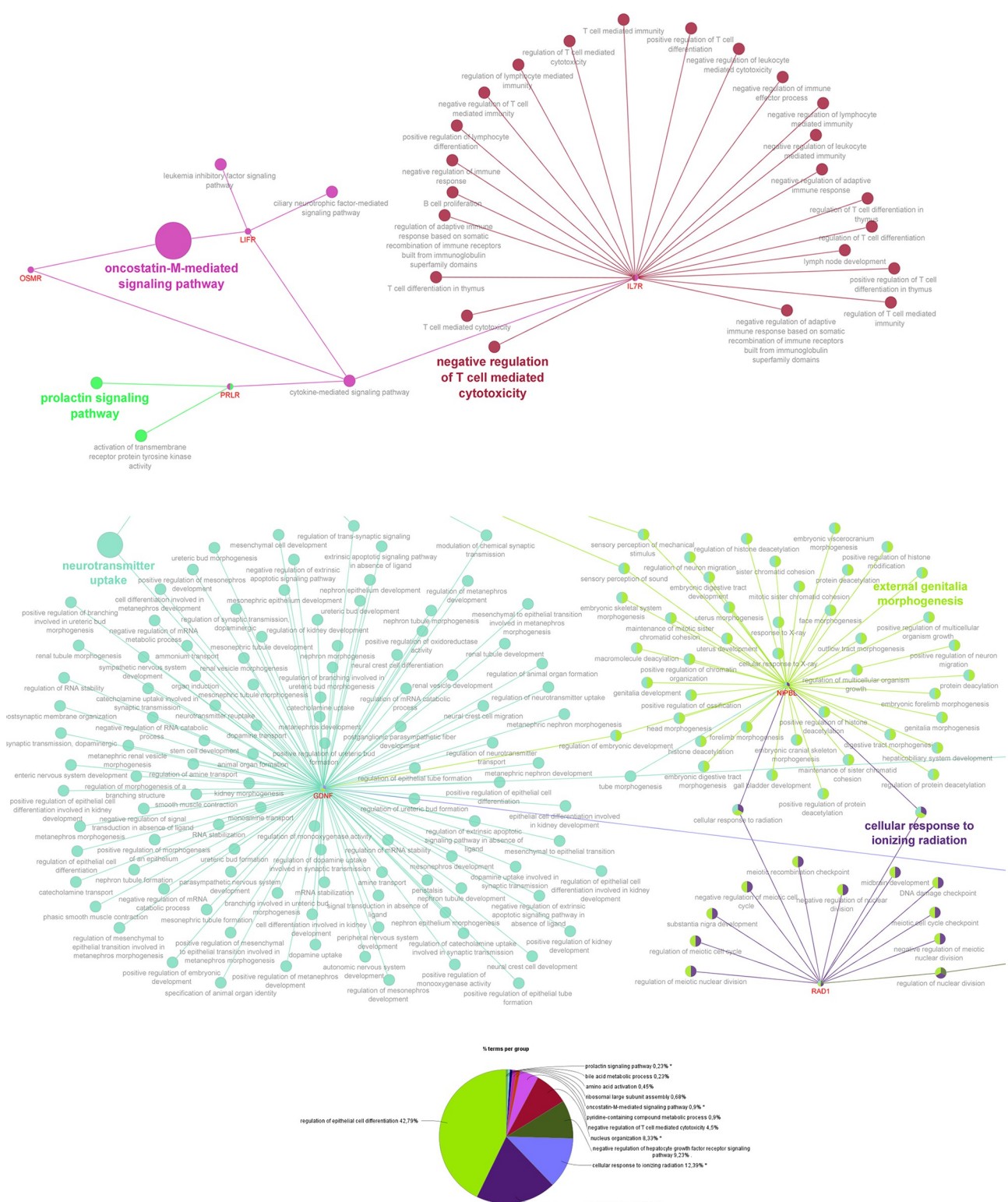

**Fig 6. Functional Network showing gene interactions in the candidate region of BTAU20 and the relationship across genes and their subnetworks.** (A) Immune system biological process; (B) Regulation of epithelial differentiation; (C) Specific gene cluster analyzes of the GO Biological Process (p≤0.01).

in a two-breed composite [25]. It seems that multiple-breeds crossbreeding resulted in a more complex genomic structure for the MONTANA TROPICAL® population.

After the development of a two-breed composite, at least 5 generations are required to form a new genomic profile [26,27]. The genomic structure of the MONTANA TROPICAL® population is much more complex due to the use of more than two breeds in its formation. Despite this, we also observed a differential genomic clustering for animals close to the fifth generation within the composite population.

The Tropical Adapted Taurine (A) biological group presented a higher proportion for late generations (ECG > 4.5). In contrast, Zebu (N) and British (B) biological groups showed a reduction in their proportions for late generations (ECG > 4.5). This is in line with the results described by Grigoletto et al. [28] where they demonstrated a switch from the use of the Zebu (N) biological group to Tropical Adapted Taurine (A) biological group in the formation of the MONTANA TROPICAL® population, which may be explained by the search for improved meat quality traits supposedly provided by taurine animals.

The results of the levels of homozygosity found in the MONTANA TROPICAL® population showed a slightly higher ROH value than those reported in another ROH study in the MONTANA TROPICAL® cattle [20]. However, the levels found in the current study are still considered low [20]. As expected, pedigree inbreeding (0.8%) underestimated inbreeding for the population when compared to $F_{ROH}$ (2%), this is probably due to flaws in the information provided by the pedigree and the inability to capture information pertaining to Mendelian sampling [15,25]. Thus, it reinforces the importance of using genomic data to better understand the real population inbreeding scenario [29]. $F_{ROH}$ is a genomic method that provides one of the most reliable measures of the true value of inbreeding in animal populations as it considers the past and recent relationship of individuals while being an accurate source of information. Subsequently, it is often recommended for calculating inbreeding in composite populations [30,31].

## Signatures of selection

The region in BTAU 20 (35399405–40329703) was the consensus selected region for both methods (ROH and iHS). This signature of selection seems to be under a strong selection pressure, as it appeared only a few generations after the composite population was formed from the crossbreeding of contrasting breeds. Therefore, the genes in this region should be very important for animals to succeed in the tropical environment where they are raised.

This region has genes encompassing functions for milk, weight, feed intake, immune, and epithelial differentiation. The gene *NIBPL* was linked to the DNA repair process (i.e, recovery from damage caused by the UV component of sun light in mammals) [32]. Also, the *NIPBL* gene is a candidate gene for growth traits for cattle [33]. Other candidate genes, *ILR7* and *LIFR*, have been identified to contribute to immune response [34–36]. Furthermore, studies have found association between the *LIFR* and *OSMR* genes and feed intake in beef cattle [37]. The *GDNF* gene is related to important functions in skin homeostasis and hair cycle control [38]. *PRLR* is extensively described as important for milk traits in dairy cows [39,40], and polymorphisms in the *PRLR* are associated with calving rates in beef cattle [41].

This same region on chromosome 20 was described as the slick locus that has been previously described in other breeds such as Senepol and Creole cattle (42,43). The slick hair coat represents very short hair in cattle [1]. This phenotype is important to facilitate the ability to dissipate heat and to maintain a stable body temperature when an animal is exposed to heat stress conditions [42]. It is unknown which gene is responsible for the slick hear phenotype, however some studies suggested different candidate genes to define this phenotype such as *GDNF* [43], *PRLR* [42], *RAI14* [44], *SKP2* [45], and *SPEF2* [45].

The slick hair coat was previously associated with the Senepol breed [42], which is one of the breeds identified in biological type A in the MONTANA TROPICAL® population composition [10]. Even after crossbreeding four different biological types, the composite population showed high homozygosity in this region related with important traits for environmental adaptation. The increasing homozygosity through ECG may be related to the high proportion of biological type A observed in late generations. Also, the MONTANA TROPICAL® selection index has a 70% weighting towards growth traits [28]. In other words, animals with the highest growth performance have a higher chance to be selected to contribute to the next generation. Individuals with a favorable interaction between genotype and environment have better productive performance. Consequently, the selection for growth traits could contribute to the selection in the candidate region identified in this work.

## Conclusion

The genomic population structure of the Montana Tropical® population is complex and shows some relationships with the founders' contribution ratio and the number of equivalent complete generations within the breed. Animals with more than 4.5 generations within the composite showed a different clustering pattern compared to previous generations. The selection pressure during crossbreeding and first generations of the composite breed left only one selected region on BTAU 20. This region harbors the slick locus and other important immune genes. Therefore, this seems to be a very important genomic region for tropical adaptation, which should be better characterized in further sequencing studies. These findings can be an opportunity for animal breeding and/or gene editing to incorporate existing genetic resources to face global climate change.

## Methods

### Animals

Genotypes (HD GGP Bovine 50K Neogen) from 1342 animals were used as the reference population for imputing the target population of 1893 genotypes using FImpute 3.0. The target population was genotyped with GGP LD BeadChip 35K Neogen (503 samples) and GGP LD BeadChip 30K Neogen (1390 samples). At the end, 3215 genotypes (HD GGP 50K Neogen, genome reference URS UCD 1.2) were used for the further analyses.

Using PLINK 1.9 [25] we applied MAF (>1%) and LD pruning (—*indep-pairwise* 50 5 0.5) filters before performing the Principal Components Analysis (PCA) and ADMIXTURE analysis. The dataset retained for these analyses had 40,634 SNP on 29 autosomal chromosomes.

For the signature of selection analyses, we kept the initial number of SNPs (51962) and did not use the MAF and LD pruning filtering, because they can cause biased ROH in medium density genotype data [42].

### Pedigree-based measures

The pedigree file was evaluated using the optiSel package in the R v3.4.2 software [46]. The mean pedigree completeness index (PCI) was 0.75, which is the harmonic mean of the pedigree completeness of the parents estimated according to the method used by MacCluer et al. [47]. The harmonic means place a higher weight on less complete paternal pedigrees, i.e. when both parents are unknow the PCI is equal 0 [48].

Animals that presented breed proportion equals to 1 for any biological group on the NABC scale were considered ancestors. The equivalent complete generation (ECG) was calculated using the equation $g = \sum (1/2)^n$ where $g$ is the number of equivalent generations and $n$ the

number of generations separating the individual from each known ancestor ECG is defined as the sum of the proportion of known ancestors over all generations traced [49]. In cattle, normally animals mate across overlapping generations. As example, a 5th generation dam can be mated with a 3th generation sire. The ECG formula accounts for this overlapping and provide a result that demonstrate the relationship of the animal with the initial crossbreeding for composite formation[49]. In our case, we considered the animal born from a crossbreeding between founder's breed that met Montana Tropical® requirements as a 1st generation. Therefore, we used ECG as indicator of how many generations each animal has inside the composite breed.

## PCA

PLINK 1.9 (https://www.cog-genomics.org/plink/1.9/) [50] was used to perform Principal Component Analysis (PCA) aiming to verify stratification and genetic distance between individuals in the population [50]. Analyses were performed with up to 10 principal components, retaining only components with eigenvalues greater than 1, following Kaiser's rule [51]. The individual eigenvectors were plotted along the different axes of a graph. The total variance explained by the main principal components was also estimated.

## ADMIXTURE

Cluster analysis based on genotyping data was performed using the unsupervised maximum likelihood method implemented in the ADMIXTURE program version 1.3.0 [52]. K values (parameter that describes the number of subpopulations that make up the total population in a dataset provided as input) of 2, 3, and 4 were used to establish the clustering patterns in the population [53]. To determine the best K value for the population, K from 2 to 20 was tested. We were not able to identify the best K as the CV error showed a continuous decline in the range evaluated S3 File.

The data was organized by classes of equivalent complete generations (ECG<2.5, 2.5<ECG<4.5 and ECG>4.5). The outputs from K = 2 to 4 were plotted using the packages ggplot2 [54], forcats [55], ggthemes [56] and patchwork [57] in the R software.

Correlation, regression and graph analyzes between PCA results, ECG, ADMIXTURE and NABC composition of the pedigree were performed using the R software with Hmisc [58], MASS [59], ggplot2 [54] and corrplot [60] packages, respectively.

## Runs of homozygosity

ROH islands were identified using the PLINK 1.9 (https://www.cog-genomics.org/plink/1.9/) [50] using the—*homozyg* function, using the following criteria [61]:

a sliding window of 50 SNPs across the genome (—*homozyg-windows-snp*);

a homozygous window overlap ratio of 0.05(—*homozyg-windows-threshold*);

a minimum length to consider a homozygous segment of 1000 kb (—*homozyg-kb*);

a maximum gap between consecutive homozygous SNPs of 1000 kb (—*homozyg-gap*);

the density of one SNP per 100 kb (—*homozyg-density*)

requirement to contain at least 20 SNPs (—*homozyg-snp*);

requirement not to allow missing genotypes and not to allow heterozygotes;

Genomic inbreeding coefficient based on ROH ($F_{ROH}$) was calculated for each animal according to McQuillan et al. [19]:

$$F_{ROH} = \frac{\sum_{j=1}^{n} L_{ROHj}}{L_{total}}$$

where $L_{ROHj}$ is the length of $ROH_j$, and $L_{total}$ is the total size of the autosomes (used the estimated value in the ARS1.2 genome assembly of 2,715,853,792 bp).

The incidence of common ROH was calculated to each group of animals classified according to generation classes. The Shapiro-wilk test of the ROH incidence results were performed to check the normality of the data and the rationality of using gaussian distribution for defining the thresholds. The threshold that defining the top 1% and 0.1% of the frequency based on gaussian distribution for each equivalent generation class was determined. The homozygous regions above the frequency threshold of each generation class of equivalent generation (5% for all generations, 4.3% for ECG<2.5, 4.9% for 2.5<ECG<4.5 and 12% for ECG>4.5) were considered as putative selected regions.

According to the length of the ROH, it is possible to estimate the number of generations traced back to the common ancestor, which generates homozygosity in that region. ROH were classified into 4 categories (1 = more than 6 generations, 2 = between 6.5 and 4.5, 3 = between 4.5 and 2.5 and 4 = less than 2.5 generations) using the equation proposed by Čurik et al. [12]: E (LIBD−H | gcA) = 100/(2 gcA), where E (LIBD−H | gcA) is the expected length of a haplotype identical per offspring (IBD) (in centiMorgans–cM), and gcA is the number of generations of the common ancestor. The conversion of the recombination rate metric to physical distance was set to 1.28 cM per Mb, which is the average of the results from Arias et al. [62] and Weng et al. [63]. Thus, for example, a 1Mb<ROH<6.5Mb probably originated from a common ancestor more than 6 generations ago, the 6.5Mb<ROH<8.7 Mb originated from a common ancestor between 6 and 4.5, the 8.7Mb<ROH<15.6Mb between 4.5 and 2.5 generations ago, and ROH more than 15.6 Mb originated from a common ancestor less than 2.5 generations ago.

## Integrated haplotype score

The iHS was performed for each autosomal SNP using the package rehh [64] in the software R. Before computing the analyzes the allele with the major frequency in the data was considered the ancestral allele.

$$iHS = \frac{ln\left(\frac{iHH_A}{iHH_D}\right) - E_p\left[ln\left(\frac{iHH_A}{iHH_D}\right)\right]}{SD_p\left[ln\left(\frac{iHH_A}{iHH_D}\right)\right]}$$

Were $iHH_A$ and $iHH_D$ represent the Integrated EHH score for ancestral and derived core alleles, respectively. $E_p\left[ln\left(\frac{iHH_A}{iHH_D}\right)\right]$ and $SD_p\left[ln\left(\frac{iHH_A}{iHH_D}\right)\right]$ are the expectation and standard deviation in terms of frequency. Windows at the top 0.1% of the empirical distribution were considered candidate regions.

## Genes, GO analysis and QTL identification

Genes in each selected region were searched in Genome Data Viewer (NCBI platform) using ARS-UCD1.3 (GCF_002263795.2). To identify traits related to genes located in each significant genomic region, a search was performed on the Animal Genome QTL Database (htt ps:// www.animalgenome.org/cgi-bin/QTLdb/EC/index).

Gene Ontology (GO) biological process network was done using Cytoscape (National Resource for Network Biology, USA, Version 3.2.1) and ClueGo (Version 2.3.5) [65]. This approach is based on a unilateral hypergeometric method and Bonferroni correction. This application provides simultaneously analyzes of 1 or more sets of genes and searches for a functional Gene Ontology (GO) term or pathways that establish relationships among genes.

## Supporting information

**S1 Table. Statistical homozygosity runs (ROH) per animal.** Standard error (SE), mean (Mb), standard deviation (SD), min (Mb), max (Mb), SROH (mean length of genome covered by ROH Mb), NROH (mean number of ROH), LROH (mean length of ROH in Mb) and FROH (inbreeding coefficient).
(PDF)

**S2 Table. All regions that was found by iHS and ROH, through different equivalent complete generations (ECG) in the consensus region of BTAU 20.**
(PDF)

**S1 File. Estimated iHS SNP in the MONTANA TROPICAL®.**
(CSV)

**S2 File. Glue GO annotation clusters of region 35399405–40329703 in the BTAU.**
(XLS)

**S3 File. CV error of different value of K.**
(PDF)

**S4 File.**
(INI)

**S1 Fig. Contribution of breed of each biological type in MONTANA TROPICAL® pedigree.**
(TIF)

**S2 Fig. Result of regression analyses between genomic inbreeding ($F_{ROH}$) coefficient and pedigree inbreeding.**
(TIF)

**S3 Fig. Result of regression analyses between the genomic inbreeding ($F_{ROH}$) coefficient and equivalent complete generations (ECG).**
(TIF)

**S4 Fig. Genomic inbreeding based on runs of homozygosity (FROH) by equivalent generation classes and by ROH length classes.** The t-test comparison results are shown at the top (ns: not significant; *$p < 0.05$; **$p < 0.01$; ***$p < 0.001$; ****$p < 0.0001$).
(TIF)

**S5 Fig. Manhattan plot IHS and ROH through ECG.** A, C, E Manhattan plot of incidence of each SNP in the ROH across individuals through different ECGs. The blue line represents the top 0.1% threshold Manhattan plot of incidence of each SNP in the ROH across individuals. The orange line represents the threshold levels of top 0.1%. of Genome-wide distribution of selection signatures detected by iHS B, D, F through different ECGs.
(TIF)

**S6 Fig. Network genes BTAU 20 (35399405–40329703).**
(TIF)

## Acknowledgments

For Animal Improvement and Biotechnology Group of the Faculty of Animal Science and Food Engineering (Pirassununga, São Paulo, Brazil) and MONTANA TROPICAL® breeders association for providing genotypic data and staff support.

## Author Contributions

**Conceptualization:** José Bento Sterman Ferraz, Tiago do Prado Paim.

**Data curation:** Camila Alves dos Santos, Joanir Pereira Eler, Elisangela Chicaroni de Mattos Oliveira, Rafael Espigolan, Gabriela Giacomini, José Bento Sterman Ferraz, Tiago do Prado Paim.

**Formal analysis:** Camila Alves dos Santos, Tiago do Prado Paim.

**Funding acquisition:** José Bento Sterman Ferraz.

**Investigation:** Camila Alves dos Santos, Joanir Pereira Eler, José Bento Sterman Ferraz, Tiago do Prado Paim.

**Methodology:** Camila Alves dos Santos, Elisangela Chicaroni de Mattos Oliveira, Rafael Espigolan.

**Project administration:** José Bento Sterman Ferraz, Tiago do Prado Paim.

**Resources:** Joanir Pereira Eler, José Bento Sterman Ferraz.

**Supervision:** Tiago do Prado Paim.

**Validation:** José Bento Sterman Ferraz.

**Visualization:** Tiago do Prado Paim.

**Writing – original draft:** Camila Alves dos Santos.

**Writing – review & editing:** Tiago do Prado Paim.

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
