## [Decision Letter · Decision Letter 0]

31 Oct 2023

PONE-D-23-33836Selective signatures in composite Montana tropical® beef cattle reveal potential genomic regions for tropical adaptationPLOS ONE

Dear Dr. Santos,

Thank you for submitting your manuscript to PLOS ONE. After careful consideration, we feel that it has merit but does not fully meet PLOS ONE’s publication criteria as it currently stands. Therefore, we invite you to submit a revised version of the manuscript that addresses the points raised during the review process.

We look forward to receiving your revised manuscript.

Kind regards,

Tzen-Yuh Chiang

Academic Editor

PLOS ONE

Journal Requirements:

   "CNPq"

Reviewers' comments:

Reviewer's Responses to Questions

**Comments to the Author**

1. Is the manuscript technically sound, and do the data support the conclusions?

Reviewer #1: Yes

Reviewer #2: Yes

2. Has the statistical analysis been performed appropriately and rigorously? 

Reviewer #1: Yes

Reviewer #2: Yes

3. Have the authors made all data underlying the findings in their manuscript fully available?

Reviewer #1: No

Reviewer #2: Yes

4. Is the manuscript presented in an intelligible fashion and written in standard English?

Reviewer #1: No

Reviewer #2: Yes

5. Review Comments to the Author

Reviewer #1: Please consider carefully all comments below:

- I want to thank you for placing figures and tables within the text! (and not at the end of it)

- but please add line numbers!!

- English: overall not too bad, but check some typos and especially the many poorly constructed sentences

- Methods must be described more precisely, with less ambiguity and providing as much as possible the exact calculations that you used

Background

----------

P10: changing climate scenario

P10: composed of

P11: arose

P11: succeeded

P11: do you have a reference/link for the registered Montana Tropical cattle? If it's a trademark/brand, please always capitalize it (including in the title)

P11: composite breed

P11: based on THE crossbreeding OF four

P11: "has growth traits as 70% of their selection index" please rephrase  growth traits account for 70% of their selection index

P12: I guess you refer to pedigree errors (you can find something on this in the literature, e.g. Leroy et al. 2012 Animal Genetics)

P12: history of population, not of pop structure

P12: rephrase to  The population frequency of ROH can be used as a tool to identify genomic regions under selection (i.e. signatures of selection)

P12: Another method to detect signals of recent selection is the integrated haplotype score ...

P12: are you sure that iHS only detects recent seelction?

Methods

-------

P27: "At the end, it was used 3215 genotypes (HD GGP 50K Neogen, genome reference URS UCD 1.2) for the further analyses presented here.": wrongly written (it was used?), please rewrite

P27: before performing

P28: signatures of selection

P28: , according

P28: what is the completeness value? How is it calculated?

P28: breed proportion

P28: what is the purpose of the ECG?

P28: "standardized variance relationship matrix"? What is this? Usually, PCA is performed on the matrix of genotypes, or on the matrix of genetic relationships calculated from the genotypes (in this latter case, we talk more precisely of PCoA: principal coordinates analysis)

P28: how did you choose the number 10? (I guess that you used the first 10 PCs ordered by decreasing eigenvalues or transformations thereof)

P29: to determine K, besides the CV error you can look also at the number of iterations needed to converge (you can see an example and explanation in rice populations from Biscarini et al. 2016, Plos One)

P30: density of ONE SNP per 100 kb

P30:LROH, LAUTO: ROH and AUTO should be suffixes here

P30: "The incidence of common ROH was transformed to generation class frequency by dividing by the number of animals from each generation class in the analysis." This is not clear, please explain better or add the calculations that you performed (preferred option)

P30: normality test of what? Which test?

P30: top 0.1% of observations in terms of what? Not clear

P31: what was the cM  Mb conversion rate that you used? 1cM = 1Mb?

P31: GO analysis

Reviewer #2: 1. Some information (for example, the description of calculated parameters, and used software) in the Results section should be moved to Materials and Methods.

2. The captions for Figure 2 are hardly readable.

3. The authors stated that FROH showed a positive correlation with inbreeding coefficient based on pedigree. I recommend providing these calculations in the Table (or as Supplementary Table). Was the correlation significant?

4. Figure 5 should be provided in a better quality.

5. The photograph of Montana Tropical cattle wil attract attention of potential readers, who are not familiar with this breed.

6. PLOS authors have the option to publish the peer review history of their article (what does this mean?). If published, this will include your full peer review and any attached files.

Reviewer #1: No

Reviewer #2: No

---

## [Author Response · Author response to Decision Letter 0]

12 Dec 2023

We appreciate the reviewer’s considerations, and we address all questions.

Reviewer #1: 

Please consider carefully all comments below:

- I want to thank you for placing figures and tables within the text! (and not at the end of it)

- but please add line numbers!!

- English: overall not too bad, but check some typos and especially the many poorly constructed sentences

- Methods must be described more precisely, with less ambiguity and providing as much as possible the exact calculations that you used

We added line numbers and performed a carefully English check. Moreover, we revised the description of methods used. 

Background

----------

P10: changing climate scenario – done 

P10: composed of – done 

P11: arose - done 

P11: succeeded - done 

P11: do you have a reference/link for the registered Montana Tropical cattle? If it's a trademark/brand, please always capitalize it (including in the title) – we added a citation of the Association website in line 52

P11: composite breed – done 

P11: based on THE crossbreeding OF four – done.

P11: "has growth traits as 70% of their selection index" please rephrase  growth traits account for 70% of their selection index – done.

P12: I guess you refer to pedigree errors (you can find something on this in the literature, e.g. Leroy et al. 2012 Animal Genetics) – done, It was a missing word before the errors. 

P12: history of population, not of pop structure – done.

P12: rephrase to  The population frequency of ROH can be used as a tool to identify genomic regions under selection (i.e. signatures of selection) - done

P12: Another method to detect signals of recent selection is the integrated haplotype score ... – done

P12: are you sure that iHS only detects recent seelction? 

According to the Voight et al. (2006), iHS method detect more recent signal of selection compared to other methods available. To avoid any misinterpretation, we removed “recent” from the statement. L:84 

Voight, B. F., Kudaravalli, S., Wen, X., & Pritchard, J. K. (2006). A map of recent positive selection in the human genome. PLoS biology, 4(3), e72.

Methods

-------

P27: "At the end, it was used 3215 genotypes (HD GGP 50K Neogen, genome reference URS UCD 1.2) for the further analyses presented here.": wrongly written (it was used?), please rewrite – done

P27: before performing – done

P28: signatures of selection – done

P28: , according – done

P28: what is the completeness value? How is it calculated? 

The completeness value is an index of pedigree completeness, which is the harmonic mean of the pedigree completeness of the parents. The relevant parameter to identify individuals with insufficient pedigree information to estimate inbreeding, however, is the PCI. This is because inbreeding can be detected only if both maternal and paternal ancestries are known. The harmonic mean ensures that the less complete paternal pedigree is weighted more heavily, so the PCI equals zero when either parent is unknown. Inbreeding coefficients can be valid despite small PCIs if the most recent founders were indeed unrelated, e.g. because they were from other breeds. L:373

MacCluer JW, Boyce AJ, Dyke B, Weitkamp LR, Pfenning DW, Parsons CJ. Inbreeding and pedigree structure in Standardbred horses. J Hered. 1983; 1;74(6):394–9. doi:10.1093/oxfordjournals.jhered.a109824. 

P28: breed proportion – done

P28: what is the purpose of the ECG? 

In the bovine mating system normally occurs the overlapping of generations. As example, a 7th generation dam can be mated with a 4th generation sire. The ECG formula accounts for this overlapping and provide a result that demonstrate the relationship of the animal with the initial crossbreeding for composite formation. In our case, we considered the animal born from a crossbreeding between founder’s breed that meet Montana requirements as a 1st generation. Therefore, we used ECG as indicator of how many generations each animal has inside the composite breed. 

P28: "standardized variance relationship matrix"? What is this? Usually, PCA is performed on the matrix of genotypes, or on the matrix of genetic relationships calculated from the genotypes (in this latter case, we talk more precisely of PCoA: principal coordinates analysis)

 We performed the PCA analysis using the genetic relationship matrix. This analysis as done using PLINK default conditions. PLINK manual (https://www.cog-genomics.org/plink/1.9/strat#pca) indicates that the command “--pca” calculate the principal components of the variance-standardized relationship matrix”. We only changed the number of retained PC according to the Kaiser’s rule. L: 385

P28: how did you choose the number 10? (I guess that you used the first 10 PCs ordered by decreasing eigenvalues or transformations thereof)

We followed Kaiser’s rule, thus we retained the number of components whose eigenvalues were greater than 1. We added this information in the text. L:388

Kaiser, H. F. (1991). Coefficient alpha for a principal component and the Kaiser-Guttman rule. Psychological reports, 68(3), 855-858.

P29: to determine K, besides the CV error you can look also at the number of iterations needed to converge (you can see an example and explanation in rice populations from Biscarini et al. 2016, Plos One)

We appreciate your suggestion. We checked the number of iterations to convergence in each run, however we still not been able to determine the best K. The minimal number of iterations was 23.3 (K=2) and the maximum was 50.7 with K = 13, then the number of iterations decreased, resulting in 40.1 iterations to converge with K=20. As mentioned by Biscarini et al. 2016 and Alexander et al. 2019, we expected that the number of iterations needed rapidly increases when the data start to support poorly the tested number of clusters (K).

P30: density of ONE SNP per 100 kb – done 

P30: LROH, LAUTO: ROH and AUTO should be suffixes here – done 

P30: "The incidence of common ROH was transformed to generation class frequency by dividing by the number of animals from each generation class in the analysis." This is not clear, please explain better or add the calculations that you performed (preferred option) L:429

The animals were grouped according to generation classes - all generations, ECG<2.5, 2.5<ECG<4.5 and ECG>4.5. The incidence of common ROH was calculated to each one of these groups. 

P30: normality test of what? Which test? Shapiro-wilk test of normality of the ROH results. L:430

P30: top 0.1% of observations in terms of what? Not clear

We calculate the threshold as top 0.1% of ROH frequency based on a gaussian distribution. In termers of the ROH frequency, this can be observed in the Manhattan plot whose the orange line represents the threshold levels of top 0.1 %.

P31: what was the cM  Mb conversion rate that you used? 1cM = 1Mb?

We calculated the average of the results obtained by Arias et al. (2009) and Weng et al. (2014) to obtain the conversion rate as stated in line 444. Both studies performed a detailed analysis of the bovine recombination map. 

P31: GO analysis - done

Reviewer #2: 

1.Some information (for example, the description of calculated parameters, and used software) in the Results section should be moved to Materials and Methods.

We removed most of these descriptions, however some of them we believed that it is necessary to explain the basic of what we have done to make the results understandable, as the material and methods is presented at the end of the paper.

2. The captions for Figure 2 are hardly readable. – We remade the plot increasing the size of the captions to get a more readable figure. 

3.The authors stated that FROH showed a positive correlation with inbreeding coefficient based on pedigree. I recommend providing these calculations in the Table (or as Supplementary Table). Was the correlation significant?

Yes, it was significant. We added these results as Supplementary Table.

4. Figure 5 should be provided in a better quality. 

 We tried our best to provide a better quality image. 

5.The photograph of Montana Tropical cattle will attract attention of potential readers, who are not familiar with this breed – We added a link to Breeder’s Association website and a photo of the animals in the text.

---

## [Decision Letter · Decision Letter 1]

24 Jan 2024

PONE-D-23-33836R1Selective signatures in composite Montana tropical® beef cattle reveal potential genomic regions for tropical adaptationPLOS ONE

Dear Dr. Santos,

Thank you for submitting your manuscript to PLOS ONE. After careful consideration, we feel that it has merit but does not fully meet PLOS ONE’s publication criteria as it currently stands. Therefore, we invite you to submit a revised version of the manuscript that addresses the points raised during the review process.

We look forward to receiving your revised manuscript.

Kind regards,

Tzen-Yuh Chiang

Academic Editor

PLOS ONE

Journal Requirements:

Reviewers' comments:

Reviewer's Responses to Questions

**Comments to the Author**

1. If the authors have adequately addressed your comments raised in a previous round of review and you feel that this manuscript is now acceptable for publication, you may indicate that here to bypass the “Comments to the Author” section, enter your conflict of interest statement in the “Confidential to Editor” section, and submit your "Accept" recommendation.

Reviewer #1: All comments have been addressed

Reviewer #2: All comments have been addressed

2. Is the manuscript technically sound, and do the data support the conclusions?

Reviewer #1: Yes

Reviewer #2: Yes

3. Has the statistical analysis been performed appropriately and rigorously? 

Reviewer #1: Yes

Reviewer #2: Yes

4. Have the authors made all data underlying the findings in their manuscript fully available?

Reviewer #1: No

Reviewer #2: Yes

5. Is the manuscript presented in an intelligible fashion and written in standard English?

Reviewer #1: Yes

Reviewer #2: Yes

6. Review Comments to the Author

Reviewer #1: Abstract

--------

- "Genomic regions related to tropical adaptability are OF paramount IMPORTANCE for animal breeding nowadays, especially ... "

- "Moreover, the genomic architecture of these traits MAY BE very RELEVANT FOR breeding programs." Why do you think that the genomic architecture of these traits is relevant for breeding programmes: pelase explain (also in the text of the Abstract)

- " ... were used to: i) characterize the population structure: ii) identify signatures of selection with complementary approaches, i.e. Integrated Haplotype Score (iHS) and Runs of Homozygosity (ROH); and iii) understand genes and traits related to selected region"

Methods

-------

- I appreciated your response to my comments on "completeness value" and "ECG": please add part of those replies to the text of the manuscript

Reviewer #2: The authors addressed comment raised during review process.

Figure 1 is missing in the revised manuscript. Please check it before the publication.

All gene names should be in italics. Check it.

7. PLOS authors have the option to publish the peer review history of their article (what does this mean?). If published, this will include your full peer review and any attached files.

Reviewer #1: No

Reviewer #2: No

---

## [Author Response · Author response to Decision Letter 1]

26 Feb 2024

We appreciate the reviewer’s considerations, and we address all questions. Hereinafter, we indicate the changes made point by point in red.

 Review Comments to the Author

Reviewer #1: 

Abstract

--------

- "Genomic regions related to tropical adaptability are OF paramount IMPORTANCE for animal breeding nowadays, especially ... " – Done

- "Moreover, the genomic architecture of these traits MAY BE very RELEVANT FOR breeding programs." Why do you think that the genomic architecture of these traits is relevant for breeding programmes: pelase explain (also in the text of the Abstract)

To know the genetic architecture of this regions is important for geneticists and breeding programs so they can make more informed decisions when selecting animals with objectives related to tropical adaptation trough hybridization for example. 

We also complement the text in the abstract. 

- " ... were used to: i) characterize the population structure: ii) identify signatures of selection with complementary approaches, i.e. Integrated Haplotype Score (iHS) and Runs of Homozygosity (ROH); and iii) understand genes and traits related to selected region" – Done

Methods

-------

- I appreciated your response to my comments on "completeness value" and "ECG": please add part of those replies to the text of the manuscript

-We added part of the explanations in the text, for completeness of the pedigree L374 – L377 and for ECG L383 – L389.

Reviewer #2: 

The authors addressed comment raised during review process.

Figure 1 is missing in the revised manuscript. Please check it before the publication. - We appreciate your comment, but in our last reviewed manuscript, we choose the option to attach the figures separately in the Editorial Manager page.

All gene names should be in italics. Check it. - Done

---

## [Decision Letter · Decision Letter 2]

4 Mar 2024

PONE-D-23-33836R2Selective signatures in composite Montana tropical® beef cattle reveal potential genomic regions for tropical adaptationPLOS ONE

Dear Dr. Santos,

Thank you for submitting your manuscript to PLOS ONE. After careful consideration, we feel that it has merit but does not fully meet PLOS ONE’s publication criteria as it currently stands. Therefore, we invite you to submit a revised version of the manuscript that addresses the points raised during the review process.

We look forward to receiving your revised manuscript.

Kind regards,

Tzen-Yuh Chiang

Academic Editor

PLOS ONE

Journal Requirements:

Reviewers' comments:

Reviewer's Responses to Questions

**Comments to the Author**

1. If the authors have adequately addressed your comments raised in a previous round of review and you feel that this manuscript is now acceptable for publication, you may indicate that here to bypass the “Comments to the Author” section, enter your conflict of interest statement in the “Confidential to Editor” section, and submit your "Accept" recommendation.

Reviewer #1: All comments have been addressed

Reviewer #2: (No Response)

2. Is the manuscript technically sound, and do the data support the conclusions?

Reviewer #1: Yes

Reviewer #2: Yes

3. Has the statistical analysis been performed appropriately and rigorously? 

Reviewer #1: Yes

Reviewer #2: Yes

4. Have the authors made all data underlying the findings in their manuscript fully available?

Reviewer #1: No

Reviewer #2: Yes

5. Is the manuscript presented in an intelligible fashion and written in standard English?

Reviewer #1: Yes

Reviewer #2: Yes

6. Review Comments to the Author

Reviewer #1: Please carefully check the English. There are some sentences which are poorly written: I provide a couple of examples below, but there are more.

L392-393: "With the harmonic mean the less complete paternal pedigree is weighted more heavily, i.e. when both parents are unknow the PCI is equal 0" This sentence is not well written: try something like "The harmonic means place a higher weight on less complete paternal pedigrees, i.e. ... "

L399-400: "In the bovine mating system normally occurs the overlapping of generations": again, not well written. Try "In cattle, normally animals mate across overlapping generations"

Reviewer #2: (No Response)

7. PLOS authors have the option to publish the peer review history of their article (what does this mean?). If published, this will include your full peer review and any attached files.

Reviewer #1: No

Reviewer #2: No

---

## [Author Response · Author response to Decision Letter 2]

16 Mar 2024

We appreciate the reviewer's considerations. We have addressed the changes and also conducted an English review of the document. We have indicated the changes made point by point in red.

Reviewer #1:

 Please carefully check the English. There are some sentences which are poorly written: I provide a couple of examples below, but there are more.

L392-393: "With the harmonic mean the less complete paternal pedigree is weighted more heavily, i.e. when both parents are unknow the PCI is equal 0" This sentence is not well written: try something like "The harmonic means place a higher weight on less complete paternal pedigrees, i.e. ... "

Done. We appreciate the suggestion.

L399-400: "In the bovine mating system normally occurs the overlapping of generations": again, not well written. Try "In cattle, normally animals mate across overlapping generations"

Done.

---

## [Decision Letter · Decision Letter 3]

25 Mar 2024

Selective signatures in composite Montana tropical® beef cattle reveal potential genomic regions for tropical adaptation

PONE-D-23-33836R3

Dear Dr. Santos,

We’re pleased to inform you that your manuscript has been judged scientifically suitable for publication and will be formally accepted for publication once it meets all outstanding technical requirements.

Kind regards,

Tzen-Yuh Chiang

Academic Editor

PLOS ONE

Additional Editor Comments (optional):

Reviewers' comments:

Reviewer's Responses to Questions

**Comments to the Author**

1. If the authors have adequately addressed your comments raised in a previous round of review and you feel that this manuscript is now acceptable for publication, you may indicate that here to bypass the “Comments to the Author” section, enter your conflict of interest statement in the “Confidential to Editor” section, and submit your "Accept" recommendation.

Reviewer #1: All comments have been addressed

Reviewer #2: All comments have been addressed

2. Is the manuscript technically sound, and do the data support the conclusions?

Reviewer #1: Yes

Reviewer #2: Yes

3. Has the statistical analysis been performed appropriately and rigorously? 

Reviewer #1: Yes

Reviewer #2: Yes

4. Have the authors made all data underlying the findings in their manuscript fully available?

Reviewer #1: No

Reviewer #2: Yes

5. Is the manuscript presented in an intelligible fashion and written in standard English?

Reviewer #1: Yes

Reviewer #2: Yes

6. Review Comments to the Author

Reviewer #1: I am satisfied with the answers to my previous comments.

Please though, do check carefully the English in which your article is written, as this is important for the quality of your publication.

Reviewer #2: (No Response)

7. PLOS authors have the option to publish the peer review history of their article (what does this mean?). If published, this will include your full peer review and any attached files.

Reviewer #1: No

Reviewer #2: No
